# RankAlign: A Ranking View of the Generator-Validator Gap in Large Language Models

**Juan Diego Rodriguez**♠* **Wenxuan Ding**♠* **Katrin Erk**♠◇ **Greg Durrett**♠

♠ Department of Computer Science ◇ Department of Linguistics
The University of Texas at Austin
{juand-r, wenxuand, katrin.erk, gdurrett}@utexas.edu

## Abstract

Although large language models (LLMs) have become more capable and accurate across many tasks, some fundamental sources of unreliability remain in their behavior. One key limitation is their inconsistency at reporting the same information when prompts are changed. In this paper, we consider the discrepancy between a model's generated answer and their own verification of that answer, the *generator-validator gap*. We define this gap in a more stringent way than prior work: we expect correlation of scores from a generator and a validator over the entire set of candidate answers, i.e., candidate completions that could possibly arise during ordinary language use without breaking Gricean norms. We show that according to this measure, a large gap exists in various settings, including question answering, lexical semantics tasks, and next-word prediction. We then propose RankAlign, a ranking-based training method, and show that it significantly closes the gap, surpassing all baseline methods. Moreover, this approach generalizes well to out-of-domain tasks and lexical items.[1]

## 1 Introduction

LLMs exhibit instability when prompted in different ways to answer the same question. One clear manifestation of this is the *generator-validator gap* (Li et al., 2024b; West et al., 2024; Hu & Frank, 2024), where a model may generate answers that it does not verify as correct, or vice versa. Resolving this inconsistency would lead to LLMs that report their underlying beliefs more consistently and do not reverse their answers when asked again, and may generally be more useful in evaluation settings (Wang et al., 2024c; Zheng et al., 2024).

Suppose that a model places probability mass over several typical answers to a question. Which of these answers should the model validate as correct? For a model to be consistent, the generator's probabilities should reflect the validator's judgments and vice versa: the generator and validator's confidences should be *correlated*.

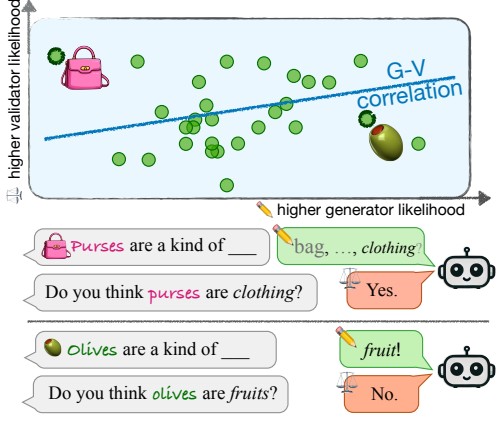

Figure 1: LLMs often have a discrepancy between generative and discriminative versions of the same task. They may generate answers that contradict their discriminative judgments or endorse responses in which the generator has low confidence.

---

*Equal contribution
[1]Our code is available at https://github.com/juand-r/rankalign.

In this paper, we introduce a new formulation of the generator-validator gap (G-V gap) that considers scores of the entire set of candidate answers at a time. Ideally, we want the model to refrain from generating responses that they disagree with discriminatively. We also expect the model to consistently assess answers even when they are less likely to generate the answers, a setting which arises when LMs are used as judges to evaluate arbitrary responses (Shi et al., 2024; Wang et al., 2024a; Li et al., 2024a) . Both desiderata will be made precise through the correlation of generator and validator scores, which can be thought of as reflecting model credences (degrees of belief).[2] It would be odd to have low credence for a proposition that one is unlikely to agree with when posed as a Yes/No question; some nuances to this issue will be discussed in §2.

We show empirically that these correlations are low for existing open-source LLMs across a range of problems, including question answering, probing for lexical semantic knowledge (hypernymy and synonymy), and next-word prediction.

We then describe a new fine-tuning approach, *RankAlign*, which uses a pairwise ranking-based loss function to align validator rankings to rankings derived from generator log probabilities. We find this strategy significantly reduces the G-V gap across models by 31.8% on average, giving substantially higher correlation between generator and validator across the population of sampled instances. Notably, it outperforms the reverse approach of aligning the generator to the validator. Moreover, our approach generalizes well to unseen tasks and to novel lexical items.

Our main contributions are: (1) a novel formulation of the generator-validator gap in terms of correlation between log-odds of generators and validators; (2) a ranking-based training objective for improving correlation between generator and validator to close the G-V gap.

## 2   Problem Formulation

Giving the answer to a question and identifying whether a proposed answer is correct are two conceptually (Campbell, 1960) and computationally (Garey & Johnson, 1979) distinct problems. In the first, generative (**generator**) case, the LM must select an answer from among a combinatorially large set of options. In the second, discriminative (**validator**) case, the solution (or set of possible solutions) is presented along with the question, and the LM must select from among a small set of options, such as indicating if an answer is correct or incorrect. LMs can give conflicting answers under generator and validator versions of the same question. This discrepancy is known as the *Generator-Validator gap* (Li et al., 2024b) or *task demand gap* between production and forced choice (Hu & Frank, 2024).

Figure 1 shows an example of how this gap can arise. When asked whether purses are clothing, Gemma-2-2B prefers `Yes` over `No`, whereas `clothing` is a less likely continuation of '`Purses are a kind of`' than is the case for other examples with a strong preference for `Yes`, ranking 32rd out of all examples. The opposite error can also occur. For example, `fruit` is an extremely likely continuation for '`Complete the sentence: Olives are a kind of`' (in fact, it is the most likely token), even though the LM generates `No` when prompted with '`Do you think olives are fruit?`'

Past work has focused on framing the G-V gap in terms of accuracy of validation decisions with respect to answers sampled from the generator (Li et al., 2024b); however, the example shows that this does not tell the whole story. This formulation fails to consider, for example, the alignment of lower-scoring (but still likely) options. We establish metrics to measure the pervasiveness of this discrepancy, and introduce a new method to help close the gap.

**Desired behavior**   It is clear that cases with high generator scores and low validator scores are unacceptable—the LM is contradicting itself (Li et al., 2024b). The reverse case is more subtle: it may be appropriate to accept propositions (high validator score) that are unlikely to be uttered (low generator score); e.g., *"Dolphins are entities."*. This is due to the

---

[2]We take the intentional stance (Dennett, 1989) and do not address the issue of whether LMs actually have beliefs.

difference between believing and uttering a proposition: it is a violation of Gricean principles (Grice, 1975) to say things that are too trivial, specific, or obscure, regardless of one's belief. For example, "Dolphins are entities" is uninformative; "Dolphins are Odonoceti" is too specific. Such extreme Gricean violations are rare in our data due to the nature of our tasks (§4.1): concept pairs are derived from commonly-ocurring categories (hypernymy), human-generated synonyms in context, predictable cloze-style completions, or simple question answering. While the completions vary in terms of typicality (Murphy, 2004), they do not do so drastically.[3]

**Setting**   We consider short-form natural language queries—questions which can be answered through a single word, entity or multi-word expression. There may be a single correct answer (e.g., TriviaQA), a set of correct answers (e.g., asking what superordinate category a concept belongs to), or a a set of answers which vary in their plausibility, which arise in more subjective tasks such as finding synonyms in context (Kremer et al., 2014), next word prediction (Paperno et al., 2016), or NLI judgments (Pavlick & Kwiatkowski, 2019).

Let the *generator prompt* $x_G$ be the sequence of tokens prompting the model to produce an answer, and denote a possible answer to the generator prompt by $y_A$. $y_A$ can be any sequence of tokens to which the *LM* can assign some probability. $x_G$ is often asking a question about a specific entity or string (e.g., 'A poodle is a kind of', or 'A synonym of chagrin is'), so we define a template function $G : z \to x_G$ to construct generator prompts concerning some $z$ of interest (e.g., poodle, chagrin).

To every generator prompt $x_G$, there corresponds an associated *validator prompt* $x_V$ which consists of a polar (Yes/No) question asking whether $y_A$ is the correct answer to $x_G$, where $y_A$ can take any candidate answer. We construct validator prompts via templates $V : (z, y_A) \to x_V$. Let $y_V$ be the token generated from the validator prompt, $y_V \sim p_{LM}(\cdot \mid V(z, y_A))$.

For example, one can probe a language model's knowledge of hypernymy via the following, where $z =$ poodle and $y_A =$ mammal:

> **Generator prompt** $x_G = G(z) =$ A poodle is a kind of
> **Answer** $y_A =$ mammal
> **Validator prompt** $x_V = V(z, y_A) =$ Is it true that a poodle is a mammal?

**Correlation of Log-Odds**   Validator prompts empirically have the property that most of the probability mass for the completion $y_V$ is concentrated on the Yes or No tokens.

We define the G-V gap so that (1) we measure agreement as a function of continuous generator/validator scores rather than binary variables, and (2) we evaluate on a range of possible completions from the generator. We operationalize these desiderata by evaluating the G-V gap through the correlation of generator and validator log-odds over a range of (generator, validator) prompts derived from a set of (question, answer) pairs (Figure 2).

In the case of a validator prompt, we wish to measure the probability mass of Yes tokens, as opposed to everything else. In practice, we observed the probability mass is concentrated on yes and no tokens, so we define the validator log-odds as:

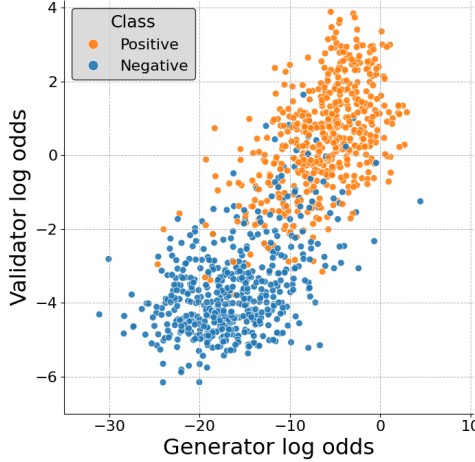

Figure 2: Generator and validator log-odds for Gemma-2-2B for hypernym prediction (Pearson $\rho = 0.76$).

---

[3]Moreover, experimental evidence (Hampton, 1998; 2007; Koriat & Sorka, 2015) shows that people's judgment of membership and typicality are in fact correlated.

$$l_V(z, y_A) = \log \left( \frac{\sum_i p_{LM}(\text{Y}_i \mid V(z, y_A))}{\sum_i p_{LM}(\text{N}_i \mid V(z, y_A))} \right) \tag{1}$$

where $\text{Y} := [\texttt{ yes, \_yes, Yes, \_Yes }]$ and $\text{N} := [\texttt{no, \_no, No, \_No }]$ are the sets of Yes and No tokens. Similarly, the log-odds of the generator is defined as:

$$l_G(z, y_A) = \log \left( \frac{p_{LM}(\text{A} \mid G(z))}{1 - p_{LM}(\text{A} \mid G(z))} \right) \tag{2}$$

where $A$ is the first token of the answer $y_A$, and $y_A$ can be any candidate answer to $x_G$.[4]

While the space of effective outputs is much larger for the generator than the validator, we can measure preferences for chosen answer continuations in both cases using log-odds.

**Log-odds Correlation ($\rho$)**   We measure the consistency between generator and validator across the range of natural (ecologically valid) answer choices $\{y_{A_i}\}_i$ via the Pearson correlation ($\rho$-**all**) of their log-odds $\{l_G(z, y_{A_i})\}_{z,i}$ and $\{l_V(z, y_{A_i})\}_{z,i}$. Since this correlation will be higher when generators and validators are both accurate (e.g., as in Figure 2) , we also evaluate the Pearson correlation between $l_G$ and $l_V$ restricted to only the set of positive examples $\mathcal{P}$ ($\rho$-**pos**), or negative examples $\mathcal{N}$ ($\rho$-**neg**).[5]

## 3   Training to Improve G-V Correlation

**RankAlign objectives**   Given a model that exhibits imperfect correlation between generator and validator, how do we go about closing this gap? When considering a single datapoint, it is difficult to calibrate via training what precise log-odds values the generator and the discriminator should return. The correlation relationship is only exhibited when looking at larger collections of points at a time. Our aim is to train a model to improve this correlation. We instantiate a simple objective to do this, which enforces positive correlation between two sampled prompts. That is, the *ranking* of the points according to the generator and discriminator must be the same.

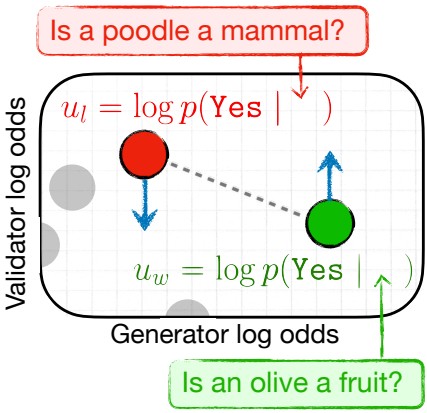

Figure 3: In RankAlign, pairwise logistic loss $\mathcal{L} = -\log(\sigma(u_w - u_l))$ is used to enforce the pair of validator log probabilities $u_w$, $u_l$ to be ordered as the generator log probabilities.

We introduce a new ranking-based method to close the G-V gap. We wish to increase the correlation between the generator and validator log-odds $l_G$ and $l_V$. Rather than optimize for this directly, we can instead encourage the LM to match the rankings of the generator with the validator, or vice versa. Whether one wishes to train an LM to have generator scores aligned with its validator scores, or validator scores aligned with its generator scores, may depend on the ultimate use case and on whether the generator or validator is more accurate.

We train the model to produce validator log probabilities that are ranked in the same way as the generator's log probabilities. Given a dataset of generator prompt-answer pairs $\{(x_{G_i}, A_i)\}_i$, we compute the generator log probabilities $\log(p_{LM}(A_i \mid x_{G_i}))$. This gives rise

---

[4]While some answers consist of multiple tokens, we observe the first token to be highly informative in most cases; this is discussed in §4.1.

[5]In some datasets (e.g., most QA datasets) only positive labels are available since all answers in the dataset are correct, in which case $\mathcal{N} = \varnothing$.

to a desired ranking between pairs of corresponding validator prompts $\{(x_{V_l}, x_{V_w})\}$ where $x_{V_l} \prec x_{V_w}$ whenever $\log(p_{LM}(A_i \mid x_{G_l})) < \log(p_{LM}(A_i \mid x_{G_w}))$.

In order to encourage the log-odds for Yes to be higher for $x_{V_w}$ than for $x_{V_l}$, we propose a ranking-based loss function:

$$\mathcal{L}_{G2V}(p_\theta) = -\mathbb{E}_{(x_w,\, x_l) \in \mathcal{D}}\Big[\log \sigma\Big(\beta[\log p_\theta(\text{Yes} \mid x_{V_w}) - \log p_\theta(\text{Yes} \mid x_{V_l})]\Big)\Big] \qquad (3)$$

where $\sigma(\cdot)$ is the sigmoid function, $\beta$ is a hyperparameter controlling the sensitivity of the preference comparisons, and $\mathcal{D}$ is the set of prompts being sampled from, described in §3. Here $p_\theta$ is the LM being trained. Note that we are not training the LM to prefer one *response* over another for a given prompt. Instead, we are training the model to assign higher likelihood to a given completion ("Yes") for one prompt compared to another.

In the case when $\beta = 1$, Eq. 3 simply minimizes the logistic cross-entropy loss over pairs $(u_l, u_w)$, $\mathcal{L} = -\log(\sigma(u_w - u_l))$ where the scores $u_w := \log p_\theta(\text{Yes} \mid x_{V_w})$ and $u_l := \log p_\theta(\text{Yes} \mid x_{V_l})$ are the log-probabilities under the validator (Figure 3). This encourages $u_w$ to be greater than $u_l$.

**RankAlign-V2G Alternative**  We also explore a variant of RankAlign, RankAlign-V2G, where we train the LM to produce log probabilities for the generator completions that match the ranking of the validator. The formulation is similar to the above, except for the slight asymmetry between the tokens over which log probabilities are computed. We rank pairs of generator prompts $\{(x_{G_l}, x_{G_w})\}$ where $x_{G_l} \prec x_{G_w}$ whenever $\log(p_{LM}(\text{Yes} \mid x_{V_l})) < \log(p_{LM}(\text{Yes} \mid x_{V_w}))$, and then use the same form of the loss as Eq. 3, except the prompts come from the generator, and the continuations are the associated answers $A_w$, $A_l$.

$$\mathcal{L}_{V2G}(p_\theta) = -\mathbb{E}_{(x_w,x_l) \in \mathcal{D}}\Big[\log \sigma\Big(\beta[\log p_\theta(A_w \mid x_{G_w}) - \log p_\theta(A_l \mid x_{G_l})]\Big)\Big] \qquad (4)$$

**RankAlign-$\alpha$**  Finally, we explore the convex combination of RankAlign and RankAlign-V2G losses:

$$\mathcal{L}_{V2G+V2G}(p_\theta) = \alpha \mathcal{L}_{G2V}(p_\theta) + (1 - \alpha)\mathcal{L}_{V2G}(p_\theta) \qquad (5)$$

in order to test whether aligning in both directions simultaneously can both reduce the G-V gap[6] and achieve high generator and validator accuracy. We set $\alpha = 0.5$ in our experiments.

**Relation to DPO**  The surface form of the ranking loss in RankAlign resembles DPO (Rafailov et al., 2023) but has fundamental differences. It pushes the LMs to establish preferences between fixed *completions* given pairs of *prompts*, rather than pairs of *completions* given fixed *prompts*. DPO also involves comparison with the reference model to prevent the LM from straying too far from the initialization (Rafailov et al., 2024). We perform an ablation by adding $p_{ref}$ to $\mathcal{L}_{G2V}$ and $\mathcal{L}_{V2G}$ in Appendix §C.2 to explore the effect of adding a reference model in our setting.

**Generator Sampling (RankAlign)**  To train with our RankAlign objective, we need to sample pairs of prompts $x_G$ and answers $A$ such that there is a clear ranking between them. We sample pairs with a minimum margin separation of $\delta$, i.e., such that $\log(p_{LM}(A_w \mid x_{G_w})) - \log(p_{LM}(A_l \mid x_{G_l})) \geq \delta$ for some $\delta > 0$.[7] Pairs are sampled uniformly at random over the dataset and filtered based on this criterion. We also report an ablation where pairs are only sampled over the set of positive examples $\mathcal{P}$ in section §5. Other hyperparameters for training are described in Appendix §A.

---

[6]Future work could consider a loss that selectively chooses the alignment direction per sample, based on whether generator or validator are more accurate.

[7]Preliminary experiments with Gemma-2-2B across all tasks showed a drop in both correlations and accuracy when setting $\delta = 0$.

# 4 Experimental Setup

## 4.1 Tasks and Datasets

We evaluate on four datasets, covering lexical relations (hypernymy and synonymy), next word prediction (cloze task), and question answering (QA). Generator and validator prompt templates for these tasks are given in Appendix §E.

**Hypernymy (THINGS)**   Hypernymy, or the IS-A relation, is a lexical relation between words and their superordinate categories (e.g., (*bee, insect*) or (*kebab, food*)). We use the dataset from Rodriguez et al. (2024) which extends the set of hyponym-hypernym pairs from THINGS (Hebart et al., 2019; 2023) with negative examples with varying degrees of conceptual similarity. The dataset is balanced between positive and negative examples.

**Lexical substitution (SWORDS)**   The lexical substitution task, or synonymy in context, is to determine whether one word can be substituted by another in a given context without altering its meaning. We use the SWORDS dataset (Lee et al., 2021), a high-quality, broad coverage dataset of (context, target word, substitutions) triples. Details on how we leverage this dataset are given in Appendix §B.1.

**Next word prediction (LAMBADA)**   We adopt the LAMBADA dataset to evaluate models' capability of predicting the next word following a given passage (Paperno et al., 2016). The dataset is designed so that final word prediction relies on the whole passage rather than just the last sentence, necessitating the understanding of the broad context.

**Question answering (TriviaQA)**   In knowledge question answering, models are tasked with providing answers using their parametric knowledge. We use the open-domain QA dataset TriviaQA (Joshi et al., 2017) and present it to models without evidence documents.

## 4.2 Metrics

In addition to our main metrics $\rho$-all, $\rho$-pos and $\rho$-neg as defined in §2, we also employ other metrics to measure the accuracy of methods on their respective tasks, and to compare against previous work (Li et al., 2024b).

**Validator performance metrics**   When the dataset contains both positive (Yes) and negative (No) labels, we can measure the validator accuracy via the area under the **ROC** curve of the validator log-odds $l_G$. This makes it possible to fairly compare methods which shift the entire distribution of log-probabilities. For the datasets with only positive labels, we use **recall** of the validator when using it to classify with a log-odds threshold of 0 (R@0).

**Generator performance metrics**   We evaluate the accuracy of the generator via the Mean Reciprocal Rank (**MRR-P**) over the ranks of the *correct* (positive) answers $A_i$ under $p_{LM}(\cdot \mid x_{G_i})$. Similarly, we evaluate the Mean Reciprocal Rank (**MRR-N**) over the ranks of the *incorrect* answers $A_i$ under $p_{LM}(\cdot \mid x_{G_i})$, i.e., over the negative examples. Smaller values are better in this case, because incorrect answers should have higher rank. We also evaluate the accuracy of the generator by calculating the rank of each answer $A_i$ under the LM with the generator prompt $x_G$, predicting positive if it falls below a rank threshold 100 and negative otherwise, and computing the accuracy over the set of $\{(x_{G_i}, A_i)\}_i^N$ pairs. We denote this **Acc@100**.

**G-V Consistency**   In order to compare against previous work, we also evaluate **G-V Consistency**, which was introduced in Li et al. (2024b). G-V Consistency measures the accuracy of the validator when presented with answers $y_A$ which are the (top) generations from the generator, $y_A \sim p_{LM}(\cdot \mid x_G)$. It measures alignment between the generator and discriminator, but only when the generator is highly confident.

### 4.3 Baselines

To evaluate the effectiveness of **RankAlign**, we compare it against the following baselines in addition to prompting the **Base** model.

**SFT**    We fine-tune the base models with both generator and validator {prompt, completion} pairs over positive examples, i.e., $\{(x_{G_i}, y_{A_i})\}$ and corresponding $\{(x_{V_i}, \text{Yes})\}$, where $(x_G, y_A) \in \mathcal{P}$. This measures the effect to which any in-domain training will increase G-V consistency by making both the generator and validator stronger.

**Consistency FT**    We evaluate the performance of the consistency fine-tuning method presented in Li et al. (2024b), which fine-tunes the model over examples where the generator and validator agree according to a binary agreement criterion. In our setting, the $y_A$s are not necessarily outputs of the generator, so we filter consistent pairs in a slightly different way. Specifically, we set thresholds $t_G = \mathbb{E}[l_G]$ and $t_V = \mathbb{E}[l_V]$ as the average generator/validator log-odds over all examples, and keep examples $\{(z, y_A)\}$ where $\mathbb{1}[l_G(z, y_A) > t_G] = \mathbb{1}[l_V(z, y_A) > t_V]$, i.e., where generator and validator agree.

**DPO**    Finally, we compare against a variant of DPO that attempts to close the G-V gap by aligning the validator with the generator based on the generator's assessment of answers, or vice versa. In **DPO-V2G**, given a validator prompt $x_V = V(z, y_A)$, we set $(y_{\text{win}}, y_{\text{lose}}) = (\text{Yes}, \text{No})$ iff $(l_G(z, y_A) > t_G$ and vice versa. Similarly, in **DPO-G2V**, we push the generator towards the validator by sampling pairs of $(y_{\text{win}}, y_{\text{lose}})$ such that $l_V(z, y_{\text{win}}) > l_V(z, y_{\text{lose}})$.

### 4.4 Models

We evaluate all methods on four models: Gemma-2-2B, Llama-3.2-3B, Llama-3.2-3B-Instruct, and Llama-3.1-8B. We test Llama-3.2-3B-Instruct on TriviaQA and lexical substitution, which are harder for non-instruction-tuned models than Hypernymy or LAMBADA. We detail the hyperparameters and other training setting in Appendix §A.

## 5 Results

Results on the Gemma-2-2B model experiments are shown in Tables 2 and 1. Full results for the other models follow broadly similar trends and are given in Appendix §C. Our main objective is to improve the generator-validator correlation, but it is also important that generator and validator performance does not degrade substantially. In particular, when aligning the validator to the generator, the generator should be minimally impacted, and vice versa.

### 5.1 In-domain Experiments

**RankAlign is effective at closing the G-V gap**    On all models and tasks, RankAlign substantially enhances the correlations between generator and validator, with an average gain of 31.8 on $\rho_{\text{all}}$. It outperforms all baselines both overall and within individual classes. For Hypernymy, RankAlign nearly closes the G-V gap, with a correlation $\rho$-all of 94.2 and per-class correlations of 87.5 and 89.0 for the positive and negative classes, respectively. On SWORDS (Table 2),

| Task | Method | $\rho$-pos | R@0 | A@100 | MRR-P |
|------|--------|-----------|-----|-------|-------|
| LAMBADA | Base | 6.1 | 90.8 | 99.3 | 79.0 |
| | SFT | 9.8 | 100 | **99.7** | **83.5** |
| | Consistency FT | 17.1 | 100 | **99.7** | 83.1 |
| | DPO V2G | -41.8 | 100 | 82.8 | 54.6 |
| | Ranking G2V | **60.0** | 67.7 | 94.8 | 52.9 |
| | Ranking V2G | 11.3 | 95.5 | 98.5 | 68.8 |
| | RankAlign-$\alpha$ | 57.2 | 2.7 | 97.3 | 74.0 |
| TriviaQA | Base | 19.4 | 63.7 | 88.6 | 52.8 |
| | SFT | 18.4 | **99.9** | 90.5 | **59.3** |
| | Consistency FT | 20.1 | **99.9** | **90.9** | **59.3** |
| | DPO V2G | 18.2 | **100** | 85.7 | 50.3 |
| | Ranking G2V | 56.8 | 39.8 | 44.5 | 9.6 |
| | Ranking V2G | 29.9 | **99.9** | 71.6 | 17.0 |
| | RankAlign-$\alpha$ | **73.1** | 75.2 | 80.0 | 36.9 |

Table 1: Performance metrics across methods for LAMBADA and TriviaQA with the Gemma-2-2B model.

| Task | Method | $\rho$-all | $\rho$-pos | $\rho$-neg | ROC | A@100 | MRR-P | MRR-N ($\downarrow$) |
|------|--------|-----------|-----------|-----------|-----|-------|-------|---------|
| Hypernym | Base | 76.4 | 54.8 | 45.1 | 97.0 | 83.7 | 19.4 | 1.6 |
| | SFT | 86.7 | 40.0 | 60.0 | **98.3** | 52.3 | **72.7** | 11.0 |
| | Consistency FT | 73.0 | 49.1 | 34.5 | 97.9 | 49.7 | 66.1 | 17.7 |
| | DPO-G2V | 77.6 | 53.6 | 49.8 | 97.0 | 85.3 | 20.1 | 1.5 |
| | DPO-V2G | 80.8 | 64.2 | 57.6 | 94.0 | 84.3 | 16.4 | 1.3 |
| | RankAlign | **94.2** | **87.5** | **89.0** | 93.5 | 83.8 | 22.1 | 1.6 |
| | RankAlign-V2G | 87.1 | 73.2 | 73.7 | 95.6 | **90.2** | 43.1 | **1.1** |
| | RankAlign-$\alpha$ | 89.3 | 82.1 | 83.1 | 93.9 | 84.6 | 41.8 | 3.7 |
| SWORDS | Base | 58.4 | 32.8 | 36.5 | 86.1 | 77.7 | 17.9 | 2.0 |
| | SFT | 58.6 | 35.2 | 36.3 | 83.8 | **79.9** | **32.5** | 3.1 |
| | Consistency FT | 55.0 | 31.6 | 31.4 | 83.7 | 77.7 | 31.0 | 3.5 |
| | DPO-G2V | 57.3 | 31.3 | 36.4 | 85.9 | 78.8 | 20.0 | 2.1 |
| | DPO-V2G | -1.8 | -6.8 | 1.1 | 50.0 | 77.5 | 20.4 | 2.5 |
| | RankAlign | 76.6 | 67.6 | 60.4 | 87.0 | 51.1 | 0.02 | 0.0 |
| | RankAlign-V2G | 47.1 | 29.6 | 33.9 | 72.4 | 79.3 | 19.2 | 2.9 |
| | RankAlign-$\alpha$ | **81.0** | **69.2** | **67.1** | **87.9** | 78.2 | 17.6 | 2.3 |

Table 2: Detailed performance metrics across tasks and methods for Gemma-2-2B on the Hypernymy and SWORDS datasets.

RankAlign increases $\rho$-all from a base 58.4 to 76.6, and it roughly doubles $\rho$-pos and $\rho$-neg (from 32.8 and 36.5 to 67.6 and 60.4, respectively). Similar improvements from RankAlign also hold for Llama-3.2-3B and Llama-3.1-8B (Tables 7, 11).

**...and has only mild degradation on task accuracy** In the case of Hypernymy, the improvement in correlations for RankAlign has little effect on model performance. Validator ROC from the base model slightly decreases (97 to 93.5 for Gemma-2, and 95.9 to 93.5 for Llama-3.2), while generator accuracy stays roughly constant, with similar trends observed for Llama. On the other hand, for SWORDS, RankAlign improves validator ROC at the expense of generator accuracy. ROC increases from 86.1 to 87 while both Acc@100 and MRR-P decrease (77.7 to 51.1 and 17.9 to 0.2, respectively). The effects on model accuracy for LAMBADA and TriviaQA are mixed. While RankAlign with Llama and Llama-Instruct has little effect on the generator accuracies of these datasets (Tables 8, 10), it causes a large drop for Gemma (a decrease in MRR-P from 79 to 52.9, and from 52.8 to 9.6 for LAMBADA and TriviaQA, respectively; Table 1). While the **SFT** baseline does not outperform RankAlign-V2G on $\rho$, it has consistently high Acc@100 values and the highest MRR-P scores across models and datasets.

**RankAlign variants** Across all tasks and models (Tables 2–10), we see that RankAlign significantly outperforms RankAlign-V2G on improving correlations, while RankAlign-$\alpha$ performs similarly. It appears that there is not enough information in the ranking of validator probabilities to successfully shift the distribution for the generator. We hypothesize that the generator's distribution may be more precisely calibrated. Base LLMs are often calibrated (Kadavath et al., 2022a), and the generator is essentially using the model as a base LLM. By contrast, the capacity to act as a discriminator is more heavily induced by alignment, and aligned LLMs may exhibit lower calibration (Zhu et al., 2023).

## 5.2 Cross-domain Experiments

Next we investigate whether our methods generalize out of domain. Ideally, one would like the G-V gap to remain small even in settings farther from the training set. We consider three cases: generalization across tasks, generalization to new lexical items, and generalization to different prompt formats.

**Generalization across tasks** We next evaluate the correlations of RankAlign in a cross-domain fashion, i.e., training them on one dataset and evaluating them on another. While correlations are naturally lower than training and evaluating on the same dataset, we

| | | → Hypernymy | | | → SWORDS | | | → LAMBADA | → TriviaQA |
| | | $\Delta\rho$-all | $\Delta\rho$-pos | $\Delta\rho$-neg | $\Delta\rho$-all | $\Delta\rho$-pos | $\Delta\rho$-neg | $\Delta\rho$-pos | $\Delta\rho$-pos |
|---|---|---|---|---|---|---|---|---|---|
| **Hypernymy** | Gemma-2 | 17.8 | 32.7 | 43.9 | 13.4 | 19.1 | 14.4 | -10.9 | 19.7 |
| | Llama-3.2 | 16.5 | 33.2 | 46.1 | 11.8 | 14.4 | 10.6 | 0.5 | 11.9 |
| **SWORDS** | Gemma-2 | 4.5 | 18.3 | 12.8 | 18.2 | 34.8 | 23.9 | 7.8 | 17.6 |
| | Llama-3.2 | 6.5 | 9.2 | 21.6 | 26.2 | 40.8 | 31.7 | 0.6 | 14.6 |
| **LAMBADA** | Gemma-2 | -6.8 | -8.7 | -12.3 | -26.5 | -15.6 | -20.9 | 57.8 | -21.4 |
| | Llama-3.2 | -0.6 | -2.8 | -0.2 | -5.7 | -5.2 | -3.6 | 45.2 | -15.5 |
| **TriviaQA** | Gemma-2 | -20.3 | -12.7 | -19.4 | -18.6 | -2.2 | -18.3 | 1.9 | 37.4 |
| | Llama-3.2 | -0.3 | 3.9 | 7.5 | 15.3 | 22.4 | 13.3 | -0.6 | 50.7 |

Table 3: Cross-dataset evaluation showing the difference in $\rho$ scores (between the RankAlign trained model and the base LM) when training on a dataset (row) and evaluating on target datasets (column). Scores greater than 0 indicate that RankAlign generalizes across tasks. Values in gray are the in-domain results derived from Tables 2, 1, 7, 8.

consider our method to have generalized if the correlations exceed those of the original base model. The deltas in correlations between the fine-tuned models and the base models across domains are given in Table 3. [8].

RankAlign generalizes well to OOD tasks when trained on Hypernymy and SWORDS in general, with large improvements over the base LM. On the other hand, LAMBADA and TriviaQA show limited generalizability to other tasks on Gemma-2, whereas they do transfer on Llama. This may be because LAMBADA and TriviaQA are more difficult and less targeted tasks, making it hard to modify the model in a systematic way. [9]

**Lexical Generalization** We evaluate whether RankAlign generalizes under varying degrees of lexical overlap between the train and test sets for the Hypernymy task, shown in Table 4. We compare the previous results from Table 2 (*Random split*) against a setting where the train and test sets have no overlap between their hypernyms $y_A$ (*No hypernym overlap*), and where there is no overlap between either hyponyms $z$ or hypernyms $y_A$ (*No overlap*). [10] We find that RankAlign generalizes well in these new settings, with only a slight decrease in correlations, while SFT continues to underperform here.

| Train/Test split type | Method | $\rho$-all | $\rho$-pos | $\rho$-neg |
|---|---|---|---|---|
| Random split | Base | 76.4 | 54.8 | 45.1 |
| | SFT | 86.7 | 40.0 | 60.0 |
| | RankAlign | **94.2** | **87.5** | **89.0** |
| No hypernym overlap | Base | 81.8 | 60.4 | 67.3 |
| | SFT | 85.0 | 68.9 | 60.1 |
| | RankAlign | **92.3** | **84.6** | **84.7** |
| No overlap | Base | 75.9 | 53.0 | 32.5 |
| | SFT | 82.2 | 4.0 | 56.4 |
| | RankAlign | **93.3** | **85.5** | **86.9** |

Table 4: Results on the Hypernymy task with varying **lexical overlap** between train and test splits.

**Generalization across prompt formats** Given that RankAlign was trained with a fixed set of generator and validator prompt templates, we evaluate whether it generalizes to closing the gap between variants of the generator and validator prompts. For the Hypernymy task, we consider prompt variants such as *"Is it the case that ..."* and *"I love _ and other"*. (Detailed prompts in Table 19.) We find that correlations drop when compared to the original in-domain prompt (Table 5), but still greatly surpass the Base model, showing that RankAlign generalizes to unseen prompts.

---

[8]Full results comparing how well methods generalize from SWORDS to Hypernymy are shown in Table 15. We note that RankAlign also has a higher correlation than the baselines in this cross-domain setting, with generator and validator accuracy similar or higher than the base model. Additional results for RankAlign-$\alpha$ can be found in Table 16

[9]To test if the lack of transfer stems from LAMBADA and TriviaQA containing only positive examples, we re-trained them with balanced positive and negative samples (details in Appendix §B.2), but observed minimal performance gains.

[10]Details on the construction of these alternative train/test splits are given in Appendix §B.3.

## 6 Related Work

**Language Model consistency** Language models are expected to be consistent in reporting their beliefs even when queried differently. Prior work has explored the instability of LMs with prompt paraphrasing (Sclar et al., 2024; Elazar et al., 2021; Moore et al., 2024), different option orders (Li et al., 2024a; Zheng et al., 2024; Ding et al., 2024), inconsistency between token probability and output (Wang et al., 2024c;b; Wen et al., 2024; Song et al., 2025), and

| Model | G–V Prompt | $\rho$-all | $\rho$-pos | $\rho$-neg |
|---|---|---|---|---|
| Gemma-2 (Base) + RankAlign | Training prompt | 76.4 | 54.8 | 45.1 |
| | | **94.2** | **87.5** | **89.0** |
| Gemma-2 (Base) + RankAlign | Variant 1 | 78.0 | 51.9 | 50. 7 |
| | | **92.0** | **84.0** | **82.9** |
| Gemma-2 (Base) + RankAlign | Variant 2 | 68.4 | 22.8 | 35.9 |
| | | **80.3** | **43.9** | **66.8** |
| Gemma-2 (Base) + RankAlign | Variant 3 | 53.6 | 21.8 | 21.0 |
| | | **76.6** | **57.0** | **64.5** |

Table 5: Correlations when evaluating RankAlign on different generator–validator prompt pairs for the Hypernymy task. Prompt variants are shown in Appendix §D). Values in gray are from Table 2, for comparison.

between logically related propositions (Li et al., 2024c; Cohen et al., 2024; Yin et al., 2024). Specifically, Li et al. (2024b) study the inconsistency where the models disagree with their own generative responses when prompted discriminatively. While they treat the generator-validator consistency as a binary agreement problem and only evaluate over candidate answers generated by the model, we provide a broader perspective, arguing that the generator and validator should align across the entire set of candidates and examples.

**Language Model as evaluators** LLMs are widely used to asses their own responses for refinement (Press et al., 2023; Wadhwa et al., 2024; Feng et al., 2024), self-alignment and scalable oversight (Sun et al., 2024; Wu et al., 2024; Jiang et al., 2024; Bowman et al., 2022), as well as acting as judges and providing nuanced insights to open-ended generation (Dubois et al., 2024; Cui et al., 2024). While LLMs-as-judges is a promising alternative to human evaluation, it is critical to understand and enhance their reliability (Shi et al., 2024; Zhou et al., 2024; Li et al., 2024d). The items to be evaluated may come from any distribution, not just those that the validator model has high confidence in. Therefore, we argue that LMs should consistently express their assessment over any candidate.

**Knowledge and belief in Language Models** It is common to attribute propositional attitudes (Nelson, 2024) such as *belief* to LMs (Jiang et al., 2020; Kadavath et al., 2022b, *i.a.*). Several studies have discussed whether LMs can have beliefs and what normative constraints (Lin, 2024) would need to be enforced for the LM's beliefs to be consistent (Hofweber et al., 2024; Hase et al., 2024; Fierro et al., 2024). The ranking perspective in this paper is also conceptually similar to Spohn's ranking theory of belief (Spohn, 2009) which assigns ordinal ranks to propositions. Finally, Gekhman et al. (2025) find that LLMs encode significantly more factual knowledge in their parameters than they express in their outputs.

## 7 Conclusion

In this paper, we present a new view of the generator-validator gap based on correlation between log-odds assigned under a generator and a validator. We describe a new method for training models to exhibit stronger correlation via a ranking loss between pairs of examples. Results show that our **RankAlign** significantly outperforms baselines with mild task accuracy degradation, is robust to prompt variations, and generalizes well to unseen data and tasks.

We believe future work can improve the generalization of these gains further and also seek to mechanistically understand the origins of these gaps, leading to new methods for more reliable language modeling. Other directions include extending the investigation to longer-form completions, which are found in LLM evaluations of summarization or long-form QA (Xu et al., 2023). On the theoretical side, it would be interesting to formulate the desired (ideal, rational) behavior relating generator and validator probabilities to a wider range of cases which violate Grice's maxims. This forces one to grapple fully with the distinction between beliefs and speech acts. One approach to this could be to take inspiration from Rational Speech Act theory (Frank & Goodman, 2012) in order to model what a rational speaker should utter (rather than believe) under generator and validator settings.

## Acknowledgments

This work was supported by NSF CAREER Award IIS-2145280, the NSF AI Institute for Foundations of Machine Learning (IFML), a grant from Open Philanthropy, and Good Systems,[11] a UT Austin Grand Challenge to develop responsible AI technologies. We thank the anonymous reviewers whose helpful feedback helped improve the work.

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

| Model | Method | Hypernym | SWORDS | TriviaQA | LAMBADA |
|---|---|---|---|---|---|
| Gemma-2-2B | Base | 0 | 0 | 0 | 0 |
| | SFT | 4497 | 1052 | 6000 | 8306 |
| | DPO G2V | 3000 | 696 | 3000 | 4153 |
| | DPO V2G | 2149 | 329 | - | - |
| | Consistency FT | 2152 | 390 | 1756 | 3156 |
| | RankAlign | 3491 | 3919 | 3753 | 779 |
| | RankAlign-V2G | 3938 | 2690 | 553 | 3039 |
| | RankAlign-$\alpha$ | 3491 | 3919 | 3203 | 3644 |
| Llama-3.2-3B | Base | 0 | 0 | 0 | 0 |
| | SFT | 4497 | 1052 | 6000 | 8306 |
| | DPO G2V | 3000 | 696 | 3000 | 4153 |
| | DPO V2G | 2149 | 329 | - | - |
| | Consistency FT | 2134 | 468 | 1884 | 2968 |
| | RankAlign | 3386 | 3146 | 2537 | 865 |
| | RankAlign-V2G | 4408 | 3762 | 986 | 1831 |
| | RankAlign-$\alpha$ | 3386 | 3146 | 3019 | 3523 |
| Llama-3.2-Instruct | Base | 0 | 0 | 0 | 0 |
| | SFT | - | 1052 | 6000 | - |
| | DPO G2V | - | 696 | 3000 | - |
| | DPO V2G | - | 329 | - | - |
| | Consistency FT | - | 530 | 2788 | - |
| | RankAlign | - | 3305 | 2902 | - |
| | RankAlign-V2G | - | 4732 | 4019 | - |
| | RankAlign-$\alpha$ | - | 3312 | 3573 | - |
| Llama-3.1-8B | Base | 0 | 0 | 0 | 0 |
| | SFT | 4497 | 1052 | 6000 | 8306 |
| | DPO G2V | 3000 | 696 | 3000 | 4153 |
| | DPO V2G | 2149 | 329 | - | - |
| | Consistency FT | 2364 | 510 | 2242 | 3214 |
| | RankAlign | 3274 | 3116 | 3242 | 3484 |
| | RankAlign-V2G | 4168 | 4505 | 3530 | 4598 |
| | RankAlign-$\alpha$ | 3274 | 3166 | 3242 | 3484 |

Table 6: Training data size for each task and model.

# A   Hyperparameters and other experimental details

**Hyperparameter details**   For **SFT**, we set the learning rate to 2e-5 and train for 1 epoch; for **DPO-V2G**, we set the learning rate to 1e-5 and train for 1 epoch, while in **DPO-G2V**, the learning rate is set to 2e-6 and train for 2 epochs according to preliminary results. In **Consistency FT**, we train with a learning rate of 2e-5 for 2 epochs.

For **RankAlign** and **RankAlign-V2G**, we use a learning rate of 1e-5 and train for 2 epochs. We use a minimum distance margin of $\delta = 2.5$ for RankAlign and $\delta = 0.15$ for RankAlign-V2G in all cases (except for Llama-3.2-3B on TriviaQA, where we found substantially better results with $\delta = 0.08$). For Hypernymy, we also experimented with $\delta = 0$, 2.5, and 5, and found 2.5 worked best.

**Dataset size for training and testing**   Our different training methods require different sampling strategies; note in particular that RankAlign-V2G and RankAlign train on sampled *pairs* of instances, allowing them to use more data in some cases than methods like SFT that train on single instances.

For SFT, we keep only positive examples and train on both generator prompts and validator prompts. For Consistency FT, we keep examples where the assessment of the generator and validator agree and train both generatively and discriminatively. For DPO and RankAlign, we sample pairs of data based on thresholds; this is described in §3 and §4.3. The specific number of examples used for training are presented in Table 6.

## B  Dataset construction details

### B.1  Dataset construction for SWORDS

The SWORDS dataset contains ratings from human annotators on how appropriate a set of replacement words are as a substitute for target word in a given context. We use this to select positive (good lexical substitutes) and negative (poor lexical substitutes) as answers $y_A$ for each context and target, by selecting the highest-ranking substitute as the positive answer and the lowest-ranked substitute as the negative answer.

We filter out examples where the target and replacement are the same word, where the replacement starts with a stopword (*the*, *be*, *a*, *in*, *yet*, *at*, *by*, *do*, *dont*, *we*, *and*, *even*, *to*, *with*), or where the replacement is a multiword expression with over 3 words. This removes roughly 4% of the data.

### B.2  Negative sampling for TriviaQA and LAMBADA

To investigate whether the lack of transfer can be attributed to the fact that LAMBADA and TriviaQA only consist of positive examples, we generate negative answers by prompting Qwen2.5-7B-Instruct. Specifically, for TriviaQA, we prompt it with "Generate an incorrect answer to the following question directly in less than 3 words.", and filter out responses that have an F1 score larger than 0.3 to reduce false negative answers. For LAMBADA, we prompt the model with "Generate an incorrect answer to the question in one word." and filter out responses that have a Jaccard-similarity greater than 0.5 to the gold completion. For each task, we then downsample to 3,000 examples consisting of balances positives and negatives and re-train with this set of data. We evaluate the resulting models on the same test set as in Table 1, which only consists of positive cases for fair comparison.

### B.3  Alternative train/test splits for Hypernymy lexical generalization experiments

In order to investigate the lexical generalization of RankAlign, we constructed alternative train/test splits, with results shown in Table 4. The *Random Split* is the default split used throughout our paper, with a random sample of 3000 training and 1000 test samples.

For the *No hypernym overlap* setting, we made sure that there were no overlaps in hypernyms between train and test sets. Since there are only 44 unique hypernyms in the dataset, we manually isolated a set of 10 (*jewelry*, *home decor*, *vehicle*, *musical instrument*, *tool*, *container*, *auto part*, *kitchen equipment*, *kitchen tool*, *garden tool*) to only use for the test set. This resulted in 3013 training instances and 1005 test instances. We verified there was minimal semantic overlap between this set and the hypernyms in the training set.

For the *No overlap* setting, we found a train/test split where there is no overlap between either hyponyms or hypernyms. This was done by viewing the set of (hyponym, hypernym) pairs as a bipartite graph, where a "no-overlap" split corresponds to a partitioning of the nodes into two disjoint sets. We used the Kernighan–Lin algorithm as implemented in the `networkx` package (v. 4.2.3) for this purpose, resulting in a train set of 2676 instances and a test set of 419 instances.

## C  Additional results

### C.1  In-domain results

**Llama-3.2-3B**    The in-domain consistency and accuracy results for experiments on Hypernymy and SWORDS with Llama-3.2-3B are given in Table 7. Results for LAMBADA and TriviaQA for Llama-3.2-3B are in Table 8.

**Llama-3.2-3B-Instruct**    Results for SWORDS with Llama-3.2-3B-Instruct are shown in Table 9 and results for TriviaQA with Llama-3.2-3B-Instruct are shown in Table 10.

| Task | Method | $\rho$-all | $\rho$-pos | $\rho$-neg | ROC | A@100 | MRR-P | MRR-N $_{(\downarrow)}$ |
|---|---|---|---|---|---|---|---|---|
| Hypernym | Base | 78.5 | 55.7 | 44.1 | 95.9 | 84.7 | 9.7 | 0.9 |
| | SFT | 88.6 | 54.5 | 61.4 | **98.0** | 51.4 | **74.6** | 12.2 |
| | Consistency FT | 79.0 | 54.3 | 43.5 | 97.4 | 49.6 | 70.0 | 16.2 |
| | DPO-G2V | 78.9 | 46.3 | 54.7 | 95.9 | 85.0 | 37.3 | 2.1 |
| | DPO-V2G | 79.4 | 48.4 | 74.9 | 91.0 | 85.0 | 8.2 | **0.8** |
| | RankAlign | 95.0 | 88.9 | **90.2** | 93.5 | 83.9 | 8.7 | 0.9 |
| | RankAlign-V2G | 93.5 | 77.7 | 82.6 | 95.9 | **90.8** | 49.6 | 2.2 |
| | RankAlign-$\alpha$ | **95.3** | **89.6** | **90.2** | 93.0 | 85.8 | 45.6 | 2.3 |
| SWORDS | Base | 53.8 | 27.0 | 32.5 | 81.6 | 76.2 | 23.6 | 3.8 |
| | SFT | 53.2 | 21.7 | 34.3 | 80.4 | 75.7 | **33.8** | 4.2 |
| | Consistency FT | 50.1 | 22.8 | 32.1 | 78.7 | 74.1 | 32.7 | 4.3 |
| | DPO-G2V | 55.2 | 27.4 | 32.9 | 81.6 | **78.4** | 27.1 | 3.9 |
| | DPO-V2G | 71.7 | 46.9 | 49.1 | 89.5 | 75.4 | 17.2 | 2.8 |
| | RankAlign | 80.0 | 67.8 | 64.2 | 89.7 | 51.6 | 0.2 | **0.0** |
| | RankAlign-V2G | 55.9 | 38.3 | 33.6 | 79.6 | 77.6 | 23.3 | 3.0 |
| | RankAlign-$\alpha$ | **84.1** | **70.4** | **70.6** | **90.5** | **78.5** | 27.0 | 3.5 |

Table 7: Performance metrics across methods for Hypernymy and SWORDS tasks with the **Llama-3.2-3B model**.

| Task | Method | $\rho$-pos | R@0 | A@100 | MRR-P |
|---|---|---|---|---|---|
| LAMBADA | Base | 9.3 | 84.6 | **99.8** | 80.5 |
| | SFT | 22.6 | 100.0 | 99.7 | **85.1** |
| | Consistency FT | 11.4 | 100.0 | 99.9 | 84.4 |
| | DPO-V2G | 39.3 | 100.0 | 99.2 | 79.2 |
| | RankAlign | **54.4** | 78.8 | 99.8 | 78.5 |
| | RankAlign-V2G | 8.9 | 93.9 | 98.7 | 57.3 |
| | RankAlign-$\alpha$ | 37.2 | 59.4 | 99.2 | **85.1** |
| TriviaQA | Base | 13.0 | 100.0 | 86.8 | 47.3 |
| | SFT | 33.6 | 100.0 | **92.5** | 63.8 |
| | Consistency FT | 37.8 | 100.0 | 92.2 | **64.0** |
| | DPO-V2G | 52.6 | 72.7 | 88.0 | 51.0 |
| | RankAlign | 70.0 | 5.0 | 86.0 | 47.6 |
| | RankAlign-V2G | 53.9 | 100.0 | 64.3 | 20.2 |
| | RankAlign-$\alpha$ | **70.8** | 13.3 | 85.5 | 40.0 |

Table 8: Performance metrics across methods for LAMBADA and TriviaQA with the **Llama-3.2-3B model**

.

**Llama-3.1-8B**  The in-domain consistency and accuracy results for experiments on Hypernymy and SWORDS with Llama-3.1-8B are given in Table 11. Results for LAMBADA and TriviaQA for Llama-3.1-8B are in Table 12.

## C.2  Comparison with DPO

The surface forms of our ranking-based loss functions bear some resemblance to DPO due to their ranking nature. Crucially, they do not incorporate the notion of a reference model, as our goal is not to adjust the likelihoods of outputs relative to a reference, but instead bring the generator and validator into better absolute alignment.

Nevertheless, to compare directly with DPO, we explore whether adding comparison with reference model $p_{ref}$ would bring performance gains in out setting. Specifically, we change the loss functions as follows:

| Task | Method | $\rho$-all | $\rho$-pos | $\rho$-neg | ROC | A@100 | MRR-P | MRR-N $_{(\downarrow)}$ |
|---|---|---|---|---|---|---|---|---|
| SWORDS | Base | 49.7 | 26.6 | 31.6 | 81.6 | 75.4 | 15.4 | 1.9 |
| | SFT | 47.8 | 27.2 | 26.5 | 78.8 | 77.9 | **24.5** | 2.7 |
| | Consistency FT | 38.1 | 23.0 | 19.2 | 72.8 | 76.3 | 22.2 | 2.7 |
| | DPO-G2V | 52.3 | 29.5 | 31.4 | 82.9 | **78.6** | 21.9 | 2.1 |
| | DPO-V2G | 50.1 | 28.3 | 22.1 | 87.6 | 63.9 | 3.1 | 0.2 |
| | RankAlign | 65.3 | 49.3 | 43.5 | 88.7 | 75.1 | 11.7 | 1.5 |
| | RankAlign-V2G | 57.0 | 37.9 | 38.2 | 82.8 | 49.1 | 0.0 | 0.0 |
| | RankAlign-$\alpha$ | **72.1** | **57.5** | **49.6** | **90.1** | 76.8 | 24.1 | 3.0 |

Table 9: Performance metrics across methods for the SWORDS task with the **Llama-3.2-3B-Instruct** model.

| Task | Method | $\rho$-pos | Recall | A@100 | MRR-P |
|---|---|---|---|---|---|
| TriviaQA | Base | 19.3 | 64.3 | 92.2 | 54.6 |
| | SFT | 34.6 | 100.0 | 94.1 | 58.2 |
| | Consistency FT | 40.4 | 100.0 | **94.2** | **58.7** |
| | DPO-V2G | 37.4 | 61.8 | 79.4 | 21.7 |
| | RankAlign | **65.4** | 0.0 | 91.3 | 52.8 |
| | RankAlign-V2G | 50.4 | 52.7 | 83.8 | 37.7 |
| | RankAlign-$\alpha$ | 37.1 | 0.0 | 78.7 | 20.3 |

Table 10: Performance metrics across methods for TriviaQA with **Llama-3.2-3B-Instruct**.

$$\mathcal{L}_{G2V}(p_\theta) = -\mathbb{E}_{(x_w, x_l) \in \mathcal{D}} \left[ \log \sigma \left( \beta \log \frac{p_\theta(\text{Yes} \mid x_{V_w})}{p_{\text{ref}}(\text{Yes} \mid x_{V_w})} - \beta \log \frac{p_\theta(\text{Yes} \mid x_{V_l})}{p_{\text{ref}}(\text{Yes} \mid x_{V_l})} \right) \right] \quad (6)$$

$$\mathcal{L}_{V2G}(p_\theta) = -\mathbb{E}_{(x_w, x_l) \in \mathcal{D}} \left[ \log \sigma \left( \beta \log \frac{p_\theta(A_w \mid x_{G_w})}{p_{\text{ref}}(A_w \mid x_{G_w})} - \beta \log \frac{p_\theta(A_l \mid x_{G_l})}{p_{\text{ref}}(G_l \mid x_{G_l})} \right) \right] \quad (7)$$

We report the results of RankAlign and RankAlign-V2G with a reference term in the loss function in Tables 13 and 14. We note that the overall effect of using a reference term in the loss function is neutral to negative.

## C.3 Cross-domain results

**Cross-task results** Additional results across models and methods when training on SWORDS and evaluating on Hypernymy are shown in Table 15. Correlations for RankAlign are lower than in the in-domain setting (Tables 2, 7), but higher than all other baselines trained on SWORDS, including Base (Table 15). In addition, RankAlign trained on SWORDS achieves a similar level of accuracy on Hypernymy as the in-domain model. Together, these results indicate that RankAlign generalizes well from the SWORDS to the Hypernymy tasks.

Results on the generalization of RankAlign-$\alpha$ across the four tasks are in Table 16 and show a similar trend to the RankAlign results in Table 3.

**Generalization across classes** We ablated the training sets for SWORDS and Hypernymy tasks to investigate whether training only on the positive examples $\mathcal{P}$ improves correlation on the negatives $\mathcal{N}$ (Table 17). RankAlign generalizes well, substantially outperforming the Base model, although falling short of RankAlign when trained on $\mathcal{P} \cup \mathcal{N}$. In comparison, the non-ablated RankAlign obtained $\rho$-neg that were 15–18 points higher for Hypernym and 7 points higher for SWORDS.

| Task | Method | ρ-all | ρ-pos | ρ-neg | ROC | A@100 | MRR-P | MRR-N (↓) |
|---|---|---|---|---|---|---|---|---|
| Hypernym | Base | 77.9 | 58.9 | 43.6 | 97.7 | 83.7 | 13.2 | 1.2 |
| | SFT | 89.3 | 58.9 | 61.8 | **98.1** | 51.5 | **73.9** | 11.7 |
| | Consistency FT | 79.8 | 51.7 | 47.1 | 97.9 | 58.9 | 69.9 | 12.9 |
| | DPO-G2V | 81.1 | 53.2 | 56.9 | 97.7 | 81.7 | 56.9 | 3.3 |
| | DPO-V2G | 76.1 | 45.2 | 76.6 | 82.8 | 82.9 | 12.7 | 1.3 |
| | RankAlign | 94.0 | **89.1** | 88.4 | 93.8 | 83.0 | 12.9 | 1.2 |
| | RankAlign-V2G | 94.8 | 77.5 | 84.1 | 97.8 | **91.7** | 35.1 | **0.5** |
| | RankAlign-α | **95.1** | 88.5 | **90.8** | 94.0 | 82.9 | 28.3 | 2.4 |
| SWORDS | Base | 71.2 | 44.5 | 48.1 | 89.7 | 77.6 | 29.9 | 3.7 |
| | SFT | 71.5 | 40.9 | 46.1 | 90.7 | 74.6 | **40.1** | 4.7 |
| | Consistency FT | 69.8 | 39.9 | 45.1 | 89.5 | 76.3 | 38.1 | 4.5 |
| | DPO-G2V | 71.0 | 44.1 | 46.7 | 90.0 | 78.7 | 30.9 | 3.6 |
| | DPO-V2G | 70.2 | 42.9 | 41.8 | 91.6 | 78.0 | 30.6 | 3.8 |
| | RankAlign | **81.4** | **69.1** | **61.3** | 90.5 | 77.5 | 29.0 | 3.6 |
| | RankAlign-V2G | 76.5 | 50.9 | 58.8 | 89.7 | 79.5 | 30.7 | **3.3** |
| | RankAlign-α | 70.2 | 42.2 | 44.8 | **90.1** | **81.0** | 26.4 | 3.4 |

Table 11: Performance metrics across methods for Hypernymy and SWORDS tasks with the **Llama-3.1-8B model**.

| Task | Method | ρ-pos | R@0 | A@100 | MRR-P |
|---|---|---|---|---|---|
| LAMBADA | Base | 0.6 | 79.6 | **99.9** | 82.5 |
| | SFT | 6.9 | 100.0 | 99.7 | **86.7** |
| | Consistency FT | 4.8 | 100.0 | 99.8 | 86.1 |
| | DPO-V2G | 62.7 | 99.3 | 99.9 | 82.3 |
| | RankAlign | **65.2** | 74.9 | 92.1 | 61.6 |
| | RankAlign-V2G | 33.5 | 38.2 | 4.1 | 0.5 |
| | RankAlign-α | 26.2 | 30.1 | 99.9 | 82.5 |
| TriviaQA | Base | 32.2 | 100.0 | 92.8 | 62.7 |
| | SFT | 36.9 | 100.0 | **95.4** | 69.9 |
| | Consistency FT | 43.7 | 100.0 | 94.9 | **72.2** |
| | DPO-V2G | 50.1 | 77.6 | 92.5 | 62.7 |
| | RankAlign | **65.2** | 0.0 | 92.1 | 61.6 |
| | RankAlign-V2G | 43.4 | 100.0 | 90.3 | 57.3 |
| | RankAlign-α | 63.6 | 2.5 | 92.6 | 62.7 |

Table 12: Performance metrics across methods for LAMBADA and TriviaQA with the **Llama-3.1-8B model**
.

| Task | Method | ρ-all | ρ-pos | ρ-neg | ROC | A@100 | MRR-P | MRR-N (↓) |
|---|---|---|---|---|---|---|---|---|
| Hypernym | RankAlign | **94.2** | **87.5** | **89.0** | **93.5** | 83.8 | **22.1** | 1.6 |
| | +ref | 93.1 | 83.0 | 86.2 | 92.5 | **84.3** | 17.8 | **1.4** |
| SWORDS | RankAlign | **76.6** | **67.6** | 60.4 | 87.0 | **51.1** | 0.02 | 0.0 |
| | +ref | 75.4 | 61.2 | 60.4 | **87.5** | 49.9 | 0.01 | 0.0 |
| LAMBADA | RankAlign | 60.0 | 60.0 | – | – | 94.8 | 52.9 | – |
| | +ref | **63.0** | **63.0** | – | – | **99.8** | **67.3** | – |
| TriviaQA | RankAlign | 56.8 | 56.8 | – | – | 44.5 | 9.6 | – |
| | +ref | **67.5** | **67.5** | – | – | **86.0** | **45.4** | – |

Table 13: Comparing RankAlign trained with and without reference terms in the loss function, for **Gemma-2-2B**.

| Task | Method | $\rho$-all | $\rho$-pos | $\rho$-neg | ROC | A@100 | MRR-P | MRR-N $_{(\downarrow)}$ |
|---|---|---|---|---|---|---|---|---|
| Hypernym | RankAlign | 95.0 | **88.9** | 90.2 | 93.5 | 83.9 | 8.7 | 0.9 |
| | +ref | 95.0 | 88.4 | 90.3 | 94.0 | 83.8 | **10.4** | 0.9 |
| SWORDS | RankAlign | 80.0 | **67.8** | 64.2 | 89.7 | 51.6 | 0.2 | 0.0 |
| | +ref | 80.5 | 64.1 | 64.8 | 90.2 | 51.3 | 0.2 | 0.0 |
| LAMBADA | RankAlign | 54.4 | 54.4 | – | – | 99.8 | 78.5 | – |
| | +ref | **60.8** | **60.8** | – | – | 99.6 | 77.9 | – |
| TriviaQA | RankAlign | **70.0** | **70.0** | – | – | 86.0 | **47.6** | – |
| | +ref | 67.4 | 67.4 | – | – | 85.5 | 33.2 | – |

Table 14: Comparing RankAlign trained with and without reference terms in the loss function, for **Llama-3.2-3B**.

| Task | Model | $\rho$-all | $\rho$-pos | $\rho$-neg | ROC | A@100 | MRR-P | MRR-N $_{(\downarrow)}$ |
|---|---|---|---|---|---|---|---|---|
| Gemma-2-2B | Base | 76.4 | 54.8 | 45.1 | 97.0 | 83.7 | 19.4 | **1.6** |
| | SFT | 75.5 | 53.9 | 43.9 | **97.2** | 83.7 | 20.7 | 1.7 |
| | Consistency FT | 76.6 | 54.3 | 47.1 | 97.1 | 83.6 | 24.0 | 1.8 |
| | DPO G2V | 76.7 | 55.5 | 45.5 | 97.0 | 84.3 | 21.0 | 1.7 |
| | DPO V2G | 59.6 | 18.1 | 40.3 | 86.2 | 83.8 | 20.8 | 1.7 |
| | RankAlign | 80.9 | 73.1 | 57.9 | 91.5 | **85.1** | 21.7 | **1.6** |
| | RankAlign-V2G | 76.5 | 54.9 | 43.2 | 96.8 | 83.6 | 28.0 | 1.7 |
| | RankAlign-$\alpha$ | **84.4** | **73.9** | **66.4** | 94.1 | 82.1 | **36.5** | 2.0 |
| Llama-3.2-3B | Base | 78.5 | 55.7 | 44.1 | 95.9 | 84.7 | 9.7 | **0.9** |
| | SFT | 78.5 | 57.1 | 45.2 | 96.1 | 83.3 | 15.5 | 1.1 |
| | Consistency FT | 78.5 | 57.0 | 45.0 | 96.1 | 83.6 | 15.5 | 1.1 |
| | DPO G2V | 79.1 | 56.3 | 45.3 | 96.0 | 84.8 | 9.6 | 0.9 |
| | DPO V2G | 82.1 | 61.2 | 56.0 | 96.1 | **85.5** | 11.9 | 1.0 |
| | RankAlign | 85.0 | 64.9 | 65.7 | 96.1 | 85.3 | 9.4 | **0.9** |
| | RankAlign-V2G | 80.7 | 57.1 | 49.5 | 96.0 | 85.4 | 14.1 | 1.0 |
| | RankAlign-$\alpha$ | **86.9** | **69.5** | **67.9** | **96.5** | **85.5** | **17.5** | 1.2 |

Table 15: Cross-domain results, SWORDS $\rightarrow$ Hypernymy, for **Gemma-2-2B** and **Llama-3.2-3B**.

| | | $\rightarrow$ Hypernymy | | | $\rightarrow$ SWORDS | | | $\rightarrow$ LAMBADA | $\rightarrow$ TriviaQA |
|---|---|---|---|---|---|---|---|---|---|
| | | $\Delta\rho$-all | $\Delta\rho$-pos | $\Delta\rho$-neg | $\Delta\rho$-all | $\Delta\rho$-pos | $\Delta\rho$-neg | $\Delta\rho$-pos | $\Delta\rho$-pos |
| **Hypernymy** | Gemma-2 | 12.9 | 27.3 | 38.0 | 16.5 | 24.7 | 16.6 | -12.9 | 13.7 |
| | Llama-3.2 | 16.8 | 33.9 | 46.1 | 13.6 | 15.2 | 13.7 | -4.4 | 25.8 |
| **SWORDS** | Gemma-2 | 8.1 | 19.0 | 21.3 | 22.6 | 36.4 | 30.6 | 2.9 | 26.0 |
| | Llama-3.2 | 8.4 | 13.8 | 23.8 | 30.3 | 43.4 | 38.1 | -6.0 | 33.3 |
| **LAMBADA** | Gemma-2 | -7.5 | -4.2 | -9.5 | -44.9 | -24.4 | -42.9 | 51.1 | -21.2 |
| | Llama-3.2 | 1.5 | 1.3 | 1.6 | -3.3 | 7.3 | -4.5 | 27.8 | 8.9 |
| **TriviaQA** | Gemma-2 | 3.7 | 6.5 | 5.2 | -1.3 | 5.1 | -3.5 | -5.7 | 53.7 |
| | Llama-3.2 | 2.2 | 4.3 | 9.8 | 16.8 | 24.7 | 17.0 | -13.8 | 57.8 |

Table 16: Cross-dataset evaluation showing the difference in $\rho$ scores (between the RankAlign-$\alpha$ trained model and the base LM) when training on a dataset (row) and evaluating on target datasets (column). Scores greater than 0 indicate that RankAlign-$\alpha$ generalizes across tasks. Values in gray are the in-domain results.

### C.4 G-V Consistency results

We evaluate G-V Consistency according to Li et al. (2024b). Specifically, we first prompt the model with generator prompts $x_G$ and set the corresponding $\hat{y}_A$ as the generator output. Then we calculate G-V Consistency as $\mathbb{E}[\mathbb{1}[y_V = \text{Yes}]]$, where $y_V \sim p_{LM}(\cdot \mid T(x_G, \hat{y}_A))$. We compare model performance in terms of **Log-odds Correlation** and **G-V Consistency** between the base model and Consistency FT-ed models.

| Task | Model | $\rho$-pos | $\rho$-neg |
|------|-------|-----------|-----------|
| **Hypernym** | Gemma-2 (Base) | 54.8 | 45.1 |
| | + RankAlign | 82.0 | 71.4 |
| | Llama-3.2 (Base) | 55.7 | 44.1 |
| | + RankAlign | 88.0 | 72.4 |
| **SWORDS** | Gemma-2 (Base) | 32.8 | 36.5 |
| | + RankAlign | 71.5 | 53.0 |
| | Llama-3.2 (Base) | 27.0 | 32.5 |
| | + RankAlign | 67.1 | 56.9 |

Table 17: Training RankAlign only on positive examples $\mathcal{P}$ generalizes to closing the gap on negative examples $\mathcal{N}$.

| Model | Task | Method | $\rho$-all | GV-consistency |
|-------|------|--------|-----------|----------------|
| Gemma-2-2B | Hypernymy | Base | 76.4 | 92.7 |
| | | Consistency FT | 73.0 | 81.2 |
| | SWORDS | Base | 58.4 | 41.5 |
| | | Consistency FT | 55.0 | 93.4 |
| | LAMBADA | Base | 6.1 | 90.5 |
| | | Consistency FT | 17.1 | 100 |
| | TriviaQA | Base | 19.4 | 80.9 |
| | | Consistency FT | 20.1 | 99.9 |
| Llama-3.2-3B | Hypernymy | Base | 78.5 | 5.0 |
| | | Consistency FT | 79.0 | 50.0 |
| | SWORDS | Base | 53.8 | 16.3 |
| | | Consistency FT | 50.1 | 56.7 |
| | LAMBADA | Base | 9.3 | 78.6 |
| | | Consistency FT | 11.4 | 100 |
| | TriviaQA | Base | 13.0 | 3.3 |
| | | Consistency FT | 37.8 | 100 |

Table 18: Comparing results on GV-consistency of base models and Consistency FT-ed models.

Results in Table 18 demonstrate that while Consistency FT does improve **G-V Consistency**, the gains in **Log-odds Correlation** are minimal or even negative. This suggests that G-V Consistency offers a limited view of consistency across the entire dataset and space of possible answers, while Log-odds Correlation provides a broader understanding of the G-V Gap.

## D   Prompt Variations

Prompts variants used in the experiments shown in Table 5 are given in Table 19.

## E   Prompt templates for experiments

We present prompt templates for each task in Table 20, 21, 22, and 24 respectively. *Optional exemplars* in validator prompts are shown to non-instruct-tuned models.

| Prompt (Generator) | Prompt (Validator) | $\rho$-all | $\rho$-pos | $\rho$-neg |
|---|---|---|---|---|
| `Complete the sentence: z are a kind of` | `Do you think z are` $y_A$`? Answer:` | 94.2 | 87.5 | 89.0 |
| `z are a kind of` | `Is it the case that z are` $y_A$`? Answer:` | 92.0 | 84.0 | 82.9 |
| `I love z and other` | `In your view, are z` $y_A$`? Answer:` | 80.3 | 43.9 | 66.8 |
| `Do you remember what our teacher used to tell us? She'd say that contrary to appearances, z are actually` | `Deep down in your bones, do you believe that z are a` $y_A$`?` | 76.6 | 57.0 | 64.5 |

Table 19: Generator and validator prompt variants and their associated cross-prompt correlations for the Hypernymy task. Here RankAlign was trained on the generator-validator prompt templates $G(z) :=$ *"Complete the sentence: z are a kind of"*, and $V(z, y_A) :=$ *"Do you think z are $y_A$? Answer:"*

| Task | Prompt |
|---|---|
| **Hypernymy** (generator) | Complete the sentence: {noun1} are a kind of |
| **Hypernymy** (validator) | Do you think bees are furniture? Answer: No

Do you think corgis are dogs? Answer: Yes

Do you think trucks are a fruit? Answer: No

Do you think robins are birds? Answer: Yes (*optional exemplars*)

Do you think {noun1} are {noun2}? Answer: |

Table 20: Generator and validator prompt templates for **Hypernym**.

| Task | Prompt |
|---|---|
| **LAMBADA** (generator) | What word is most likely to come next in the following text?
Text: {text} |
| **LAMBADA** (validator) | Is the word "anyway" the most likely word to come next in the following text?
Text: "She gently takes him by his shoulders, forcing him to face her, and she adjusts the angle of his tie the way she might straighten a picture on the wall. "I'm sure I don't need to tell you how important this gala is."

"You don't, but you will"

Answer: Yes (*optional exemplar*)

Is the word "{completion}" the most likely word to come next in the following text? Text: {text}

Answer: |

Table 21: Generator and validator prompt templates for **LAMBADA**.

| Task | Prompt |
|------|--------|
| **SWORDS** (generator) | Notice the word "{target}" used in the context: "{context}". In this context, the word "{target}" is synonymous with " |
| **SWORDS** (validator) | Determine whether the word in context can be replaced by another word or expression without changing the meaning of the sentence.

Notice the word "artists" used in the context: "Many painters, sculptors, and other *artists* were inspired by Duchamp.". In this context, is "artists" synonymous with "character"? Answer: No

Notice the word "happen" used in the context: "I could free Tasha. If I did, one of three things would *happen*. Most likely: she would be meat..." In this context, is "happen" synonymous with "transpire"? Answer: Yes (*optional exemplars*)

Notice the word "{target}" used in the context: "{context}". In this context, is "{target}" synonymous with "{replacement}"? Answer: |

Table 22: Generator and validator prompt templates for **SWORDS**.

| Task | Prompt |
|------|--------|
| **TriviaQA** (generator) | Question: {question}

Answer: |
| **TriviaQA** (validator) | Is the correct answer to the question "What kind of song is a Brindisi?" given by "drinking song"? Answer Yes or No: Yes (*optional exemplar*)

Is the correct answer to the question "{question}" given by "{answer}"? Answer Yes or No: |

Table 23: Generator and validator prompt templates for **TriviaQA**.

| Task | Prompt |
|------|--------|
| **TriviaQA** (generator) | Question: {question}

Answer: |
| **TriviaQA** (validator) | Is the correct answer to the question "What kind of song is a Brindisi?" given by "drinking song"? Answer Yes or No: Yes (*optional exemplar*)

Is the correct answer to the question "{question}" given by "{answer}"? Answer Yes or No: |

Table 24: Generator and validator prompt templates for **TriviaQA**.

