# OpenReview forum: "RankAlign: A Ranking View of the Generator-Validator Gap in Large Language Models"
_colmweb.org/COLM/2025/Conference — COLM 2025_

### Official Review · Reviewer_JEXo · 2025-05-10

**Rating:** 7
**Confidence:** 3
**Ethics Flag:** 1

**Summary:**

RankAlign is a method of training a language model (LM) to increase the consistency of the model between generator and validator variations of the same task.  For example for a "hypernymy" task, a generator would be "A horse is a type of ____" (looking for mammal) while a validator woudl be "Is a horse a type of mammal?" (looking for "Yes").  Note that a high score for the first should imply a high score for the second, and to some degree vice versa.  The authors experiment with 4 such tasks, and experiment on how well their method improves the consistency (called the G-V) gap of several small/mid-sized LMs: Gemma-2-2B, Llama-3.2B, and Llama-3.2B-Instruct.  Their primary method adjusts the validator's log probabilities to be ranked in the sway way as the geneator's, and they also experiment with adjusting the other direction, and compare these to baselines such as Supervised Fine-tuning (SFT), Consistency Fine-tuning (Li et al., 2024), and Direct Preference Optimization (DPO).  Their technique often leads to the highest Pearsen correlation between generator and validator probabilities, while usually not losing much in terms of accuracy of the generator on the task.  They also try some generalization (cross-domain) experiments, for which their technique hold up in some cases (e.g., unrelated words, different forms of prompts) and not as well for others (training on one task only sometimes leads to improvement in correlation on a different task).

Overall, the paper is an interesting experiment and an attempt to tackle one challenge with LMs.  The authors are reasonably successful on a small scale, and it may point the way to larger scale or scope techniques.

The limited scope is the main weakness of this paper.  The models used for the experiments are quite on the small end of LMs being used today.  At least from this paper, it is unclear if this G-V gap is really a problem with all models or just the smallest of them.  And the scope of tasks (4) is also quite limited.  This weakness wouldn't be an issue if the cross-task generalization was high, but since this is a weak point of their technique, it is hard to know how useful this kind of training will be for general use.  Overall, I do think it is an interesting step, but unless the scale and scope issues are tackled by future work, it will remain merely interesting.

**Reasons To Accept:**

- interesting topic/relevant problem
- reasonable experiments
- method seems to work for at least a limited scope

**Reasons To Reject:**

- experiments are all small scale, hard to know if problems (or solution) will generalize to larger models
- method shows limited ability to generalize beyond the scope of the training task

---

> ### Author Response · Authors · 2025-06-02
>
> We thank the reviewer for helpful comments.
>
> Here we address the reviewer’s two stated weaknesses:
>
> 1. “The method shows limited generalizability beyond the stated task”. Here we would like to point out that on two (out of three) OOD tasks, SWORDS and Hypernymy generalize quite well (the first two rows of Table 4). In particular, training on lexical-level tasks generalizes to complex and practically useful question answering tasks (TriviaQA). We are currently expanding the set of tasks and our understanding of generalization of the G-V gap for a forthcoming paper.
>
> 2. “Hard to know if problems (or solution) will generalize to larger models”. We chose some of the best performing “small” models; note that Llama-3.2-3B is the largest of the two text-only Llama-3.2 models, and actually outperforms some larger models (e.g., Llama-2 13B). The G-V gap is also still a problem for larger models, e.g., for Gemma-2-9B (with correlations of 74, 38, 12 for SWORDS,  TriviaQA, and LAMBADA, respectively) and for Llama-3-8B-Instruct (with correlations of 63, 31, and 10 for SWORDS, TriviaQA and LAMBADA, respectively).

---

> > ### Comment · Reviewer_JEXo · 2025-06-09
> >
> > Thank you for your comments.
> >
> > Regarding generalization, perhaps I am not understanding table 4, but it looks to me like Hypernymy and SWORDS generalized only to a limited extent (not nearly as much as training on the task's own data).  This generalization only really applied to the two tasks closest to each other.  While a start, this does not imply you've solve the G-V gap problem in general -- we cannot assume that this (limited) result carries over to other tasks or situations.  I look forward to your upcoming paper which will hopefully address this to a greater extent.

---

### Official Review · Reviewer_p48J · 2025-05-12

**Rating:** 3
**Confidence:** 3
**Ethics Flag:** 1

**Summary:**

The generator-validator gap is a new type of language model inconsistency that is particularly important now that LLMs are starting to be used for judges. This paper propses a new metric, the correlation between a generator and it's validator, in order to quantify this type of inconsistency. The are able to ameliorate poorly aligned models by introducing a ranking-based loss that enforces alignment. While this approach results in degredation of accuracy and other measures of quality, they show that their method is able to close the gap (as quanitifed by their method but not previous methods).

The work is self is clear and the datasets and main experimental conditions are well chosen. The loss is novel and interested attempt to solve for problems introduced and initially attempted by [1]. However, the metric is conceptually quite confusing and generally not well validated which makes the work reads like: "We have introduced our own metric which we have improved on" this is concerning especially given the results in C.4 which seems to indicate that this metric is negatively correlated with previous metrics in the literature.

Aside from the problems in the metrics, this work has limited significance in the field of LLM consistency because the setting isn't connected to paraphrase inconsistency, inconsistency in sampling, or calibration literature on uncertainty quantification. It really isn't made clear why optimizing for G-V correlation as it is formulated in this paper is a good thing (since it introduces degenerate cases such as encouraging completion of low perplexity but correct words i.e. A poodle is a kind of thing or A poodle is a kind of mammal, which should not come before A poodle is a kind of dog.

[1] Li, X. L., Shrivastava, V., Li, S., Hashimoto, T., & Liang, P. (2023). Benchmarking and improving generator-validator consistency of language models.

**Reasons To Accept:**

The RankAlign objective is an interesting approach that works quite well for its stated purpose.  The experimental setup is generally well done (aside from the metrics validation). Supposing that we agreed the GV correlation metric was well validated and something the community wanted, then I would recommend this paper for acceptance.

**Reasons To Reject:**

## (1) Metric Motivation and Formulation

This metric is conceptually flawed, using figure 1 we see naturally that "Olivers are a kind of fruit" and "Do you think olives are fruits? No" are inconsistent and we should not expect a discrepency. The second example "Poodles are a kind of dog" and "Do you think poodles are mammals? Yes", Is not inconsistent at all. There is no problem with that second example - the generator and the validator are already aligned in that case. I can understand the desire to formulate a metric unrelated to accuracy (lines 76 to 80) but this would apply for cases like: "Olivers are a kind of dog" and "Do you think olives are dogs? Yes" not for the poodle example. Further it doesn't seem like justification for why the poodle example should be included in the metric. This conceptual flaw likely explains why RankAlign results in degraded accuracy and isn't well validated (Appendix C.4). My recommendation for this is to reformulate the metric so it only captures the olive examples I provided but not the poodle examples. This isn't something in my view that can be resolved during the rebuttals but requires major revisions and resubmission.

## (2) Metric and Method Validation

Supposing the metric was fine conceptually, the metric isn't well validated. The only validation is in Table 15, where we see the metric does not align with previous approaches. The authors need to provide experiments showing that correlation corresponds to previous metrics of GV consistency.

Since the results are all presented in the context of this metric and task accuracy alone, the paper reads as if the authors constructed a metric that they were able to improve instead of reading as a genuine improvement in generator-validator gap improvements. This is clear in Table 15, where the metric doesn't seem to align with how previous literature thought of GV-consistency. RankAlign itself is thus not validated, since alternative consistency measures are not provided. As above, I don't believe either of these can be addressed during a rebuttal period but I would be happy to provide further guidance and suggestion during the rebuttal period.

## (3) Significance

The approach to sampling is not mentioned in the paper at all. I assume default configurations of deterministic sampling is used.
Since this paper is about consistency, I would have expected for multiple seeds to have run with a random sampling algorithm such as nucleus sampling. I think this paper would be improved if this was added - It is hard to tell how the GV gap of each model varies with sampling.  Similarly, paraphrase variation is quite common in consistency literature yet is not provided - adding experiments over paraphrase varation would futher ehance the significance of this work to consistency research.  Without these experiments, metric isn't as interesting and useful to the community as it could be.

---

> ### Author Response · Authors · 2025-06-02
>
> We thank the reviewer for their feedback which we will use to clarify certain points in our paper.  We appreciate the reviewer’s comments that the RankAlign objective is interesting and works well at reducing the G-V gap.
>
> We address the three specific critiques below:
>
> 1. “The metric is conceptually flawed”
>
> We feel there is a misunderstanding here. Briefly, the reviewer is saying that:
>
>
> (A) Our metric is measuring the inconsistency of examples like “Poodles are a kind of__” (LM: “dog”) and “Do you think poodles are mammals?” (LM: “Yes”), which are in fact already consistent.
>
> and
>
>
> (B) Our metric does not, but should, account for cases like “Olives are a kind of __” (LM: “dog”) and "Do you think olives are dogs? (LM: “Yes”) (e.g., “My recommendation for this is to reformulate the metric so it only captures the olive examples I provided but not the poodle examples.”)
>
> For point (A), we agree that saying poodles are dogs and asserting to poodles being mammals is consistent, if we treat consistency as a binary metric following previous work (Li et al., 2023). However, this is not what we were trying to illustrate in Figure 1. In the figure, “dog” was being shown as the most likely next token, whereas our focus is on the token “mammal”, which is ranked much lower (meaning much lower generator probability as shown in the figure).  A very low probability for “mammal” is inconsistent with generating “Yes” to “Do you think poodles are mammals?” with high validator confidence, because the validator is asserting something that the LM is unlikely to ever generate. In our view, the high validator score should imply a higher generator probability as well.
>
> For point (B), we argue that our metric should actually *not* treat this case as inconsistent. In this case if the LM strongly prefers both “dog” (generator) and “Yes” (validator) then they actually agree (are consistent), even though both are wrong.
>
>
> 2. “Metric and Method Validation”, i.e, we should provide experiments showing that our evaluation of the G-V gap correlates with previous metrics like G-V consistency.
>
> We disagree with this point. Our goal in this work is to introduce a new measure of G-V consistency, which we believe is conceptually useful, and improve on that. We argue that validating our metric based on comparing to past notions is not appropriate. G-V consistency from (Li et al., 2023) only measures completions where the generator has high likelihood (where the next token(s) are decoded greedily) and the validator disagrees with those (e.g., the “olive” example in Figure 1). However, our metric considers the entire spectrum of generator likelihoods and much broader set of candidate completions. In essence, we argue for a more holistic notion of consistency, and our method is designed to improve on this holistic notion.
>
> We included Table 15 in the Appendix mainly to show the reader that our implementation of Consistency FT does improve G-V Consistency, in line with (Li et al., 2023). Our intention was not to comment on the relation between G-V consistency and log-odds correlation, and we will make that clearer in the camera ready version of the paper.
>
> 3. “Significance”
>
> (a) Sampling: we did not address sampling from the Language Model because we did not need LM sampling in our paper given how we defined the G-V gap. We evaluate the G-V gap with respect to a given dataset of (generator, validator) prompts and answers (e.g., “an olive is a __” and “fruit”). We then look at the log-odds for “Yes” (for the validator) or for the answer (for the generator), so we do not have to sample from the LM.  We argue that this actually makes our metric *more* useful to the community, because it looks at a wide range of possible outputs, restricted only by the dataset under consideration, and so with RankAlign one can focus on evaluating the LM, rather than the decoding method.
>
> (b) Paraphrase variation: consistency over paraphrase variations is a different notion of consistency than our G-V consistency defined here. Relatedly, we did evaluate generalization of our method under paraphrase variation in our paper; see section 5.2, under “Generalization across prompt formats”. However, since the main goal in this paper is to reduce the G-V gap, we see further experiments with paraphrase consistency as outside the scope of the current work.

---

> > ### Comment · Reviewer_p48J · 2025-06-10
> > **Thank you for the response keeping the score.**
> >
> > I appreciate the response, I am going to keep the score for several reasons:
> > (1) Thanks for the clarifications on this, according to the paper I am still reading (A) as being considered as inconsistent under the metric.
> >
> > For the AC, here is the crux of our disagreement:
> > > A very low probability for “mammal” is inconsistent with generating “Yes” to “Do you think poodles are mammals?” with high validator confidence, because the validator is asserting something that the LM is unlikely to ever generate. In our view, the high validator score should imply a higher generator probability as well.
> >
> > I don't agree that this claim is true and that I don't think that "In our view, the high validator score should imply a higher generator probability as well" is a good thing to do (which is clearly seen with the degredation introduced by the alignment method). Because of this I don't feel as though this metric is conceptually valid.
> >
> > (2) I understand the point where direct metric comparison *could* introduce confusion and hence the authors feeling like its not appropriate, but for empirical validity, introduction of new metrics requires a robust study and comparison with past metrics even if its to show why they result in different scores, I don't see how the paper can be accepted without this.
> >
> > For (3) Understood on both points and sorry for missing 5.2 that is my bad. The argument about sampling resonates with me, making that clearer in the paper might be a good idea.

---

> > > ### Author Response · Authors · 2025-06-11
> > >
> > > We understand and agree with your statement about the crux of the disagreement. Nevertheless, we assert that our definition is meaningful.
> > >
> > > We take the likelihood of the completion to be a proxy for the credence of the model—how likely the model is to believe the given proposition, under the caveat that other factors such as typicality may play a role. It is then inconsistent to assert “Yes” to a statement that one does not believe to be True; this is known in epistemology as the Knowledge Norm of Assertion (due to Tim Williamson).
> > >
> > > There is a large class of prompts where this notion is meaningful in practice. For example, in a dataset like TriviaQA, the model should only provide a valid answer to the question, and we see no reason why for a closed-form QA setting, the generator and validator shouldn’t be aligned in the way we are suggesting. For open-ended prompts like “List a random mammal” (which we didn’t study in this work), we argue that our notion applies as well: the model should place a uniform distribution over everything its validator considers to be a mammal.
> > >
> > > When we have an open-ended question that asks for typical responses, like “What are some common ways to do [task]?“, we agree that there is a tail of valid responses where the generator may assign them lower probability. We plan to clarify this motivational point more clearly in any future version. Thanks for engaging in the discussion and providing feedback on our work!

---

### Official Review · Reviewer_Zso5 · 2025-05-13

**Rating:** 6
**Confidence:** 2
**Ethics Flag:** 1

**Summary:**

This work explores the generator-validator gap – where a model’s verification of an answer and generation of an answer to a question -- do not agree and proposes a new way to both formulate and mitigate the gap. Instead of the binary case where the gap is defined as the validator not picking the generator’s answer, this work instead operates in the continuous space where the generator and validator scores should correlate over a set of candidate answers. To do this, this paper sets up a simple case where examples are sentences with single word completions for the generator (e.g. *“A poodle is a kind of _”*) and corresponding yes/no questions for the validator (e.g. *“Is a poodle a kind of _?”*). The gap is measured using the correlation between the generator and validator’s next token log-odds.

In addition, this work proposes a new way to reduce the gap through finetuning called RankAlign. This approach aligns the validator’s candidate rankings with candidates’ log-likelihoods from the generator. They also test the reverse case, RankAlign-V2G, where the generator is trained to align with the validator’s rankings. Overall, RankAlign shows a significant improvement in the correlation between generator and validator without impacting task performance. These improvements are demonstrated by training in-domain and transferring cross-domain.

**Questions To Authors:**

- Do you think RankAlign can support thinking/reasoning models? They're useful LLM-as-judges but their step-by-step outputs don't seem to fit into your current framework.

**Reasons To Accept:**

- The overall formulation of the generator-validator gap is well laid out. The idea of exploring correlations across large sets of examples/candidate answers instead of a binary score seems very reasonable for better model generalizability.
- RankAlign is fairly straightforward and has clear impacts in reducing the gap without compromising performance. This seems to hold across different language models as well. (I'm curious if how these results would scale to a larger model >3B).
- The evaluation tasks chosen are pretty well varied. They test different aspects of semantic understanding (THINGS, SWORDS, LAMBADA) and there's the knowledge dataset (TriviaQA). The difference in correlation across tasks also fits with what I would intuitively expect, such as TriviaQA+LAMBADA being the more challenging and therefore having the lower correlations.

**Reasons To Reject:**

- The evaluation data is somewhat limited. The input prompts and target generations are short-form. It’s not clear how well RankAlign would generalize to models used as judges with longer, more open-ended input prompts. These freeform cases would be the likely use case for LLM-as-judge.

---

> ### Author Response · Authors · 2025-06-02
>
> We thank the reviewer for their detailed feedback on our paper.
>
> > The evaluation data is somewhat limited.
>
> We selected 4 tasks that we felt showcased a range of inference types. These include lexical semantics tasks (hypernymy, synonymy), a task involving world knowledge (TriviaQA), and next word prediction (LAMBADA).   SWORDS has moderately long input passages. We agree that extending our evaluation to longer outputs would be useful. However, paraphrase variation (different ways of expressing the same content) makes it difficult to concretely evaluate an LLM for the G-V consistency property as we’ve defined it here. Our work represents a first step towards formulating and evaluating the G-V gap in this way, and we believe that future work can extend it with a focus on long-form outputs.
>
> > Do you think RankAlign can support thinking/reasoning models?
>
> To evaluate the G-V gap for a model with CoT, in principle we would need to marginalize over the CoT generation to find the probability of generating an answer given the input prompt. This could be approximated by sampling, but the quality of this approximation is unclear and we don’t see an easy way to conduct rigorous evaluation in this setting. We think this would also be a great step for future work!

---

### Official Review · Reviewer_R1Jh · 2025-05-13

**Rating:** 6
**Confidence:** 3
**Ethics Flag:** 1

**Summary:**

This paper addresses the generator-validator gap (G-V gap) in large language models (LLMs), where models give inconsistent responses when prompted in different ways to answer the same question. The authors propose a new formulation of this gap based on the correlation between log-odds scores from a model's generator (when creating answers) and validator (when verifying answers) modes across the entire set of candidate answers.

**Questions To Authors:**

None

**Reasons To Accept:**

1. The paper presents a more stringent and comprehensive definition of the generator-validator gap, moving beyond the binary agreement considered in previous work to analyze correlation across the entire distribution of possible answers.
2. RankAlign shows clear improvements over all baseline methods, with an average gain of 31.8% in generator-validator correlation across all tasks and models tested. The authors conduct extensive evaluations across different models (Gemma-2-2B, Llama-3.2-3B, Llama-3.2-3B-Instruct) and four diverse tasks, providing a comprehensive analysis of their method's effectiveness.
3. RankAlign demonstrates impressive generalization to unseen lexical items, different prompt formats, and even entirely different tasks, indicating that the method captures a fundamental aspect of model behavior.
4. The ranking-based training approach is well-motivated and explained, with ablation studies and comparisons to related methods like DPO that help clarify its unique contributions.

**Reasons To Reject:**

1. While RankAlign significantly improves correlation, this sometimes comes at the cost of task accuracy. For example, on TriviaQA with Gemma-2, MRR-P drops from 52.8 to 9.6, and on SWORDS, Acc@100 decreases from 77.7 to 51.1. These trade-offs are acknowledged but not fully addressed.
2. The method's effectiveness varies considerably across models. For instance, RankAlign causes significant performance drops on Gemma-2 for LAMBADA and TriviaQA, but much smaller effects on Llama models. This variability is not fully explained.

---

> ### Author Response · Authors · 2025-06-02
>
> We thank the reviewer for their detailed feedback.
>
> Here we address the two weaknesses identified by the reviewer:
>
> 1. RankAlign does sometimes cause a drop of generator accuracy. We note however, that validator accuracy is hardly affected for Hypernymy and SWORDS. We believe our method shows promise for certain focused settings, and future work can build on it to develop methods that maintain or improve both generator and validator accuracy as well.
>
> 2. We have a few hypotheses for the differing performance across models. Different models have different base capabilities, as shown by their different base performance on the tasks and how they benefit to different extents from SFT. The validation threshold of each model is also different, as shown by the different R@0 numbers. Intuitively, these factors have an impact, but their relationship is complex and we do not have a simple, evidence-grounded story to convey, so we leave this for future work. Also in the future we can apply our method to other open LMs to determine how characteristic the drop in accuracy is for Gemma-2-2B.

---

> > ### Comment · Reviewer_R1Jh · 2025-06-10
> >
> > Thanks for your clarification. I am still a little concerned about the effectiveness.  I tend to keep the score unchanged.

---

### Decision · Program_Chairs · 2025-07-08

**Decision:**

Accept

**Comment:**

This paper aims to measure and reduce the generator-validator gap in LLMs via a ranking-based fine-tuning method called RankAlign. The paper is clearly written with thorough experiments  showing significant gains at closing the gap. However, the majority of reviewers had concerns about generalization to larger models (>4B) and different tasks, as the results were not very conclusive on how much cross-task generalization exists. Furthermore, there were also concerns about the method causing performance drops on certain model/task pairs (e.g., TriviaQA and LAMBADA for Gemma-2). Overall, the paper is well executed and feels like a good first step towards measuring and closing the G-V gap, but the proposed method needs more empirical validation to argue convincingly that it doesn't harm performance across larger open-weight models.